Fine motor deficits in reading disability and language impairment: same or different?

Brookman Annie
McDonald Sarah
McDonald David
Bishop Dorothy V.M. dorothy.bishop@psy.ox.ac.uk
Department of Experimental Psychology, University of Oxford , Oxford , UK
McArthur Genevieve
Electronic publication date: 2013 Nov 28
Publication date: 2013
Volume: 1
Electronic Location ID: e217
Received 2013 Jul 25; Accepted 2013 Nov 6
Copyright: © 2013 Brookman et al.
Copyright year: 2013
Copyright holder: Brookman et al.
License: This is an open access article distributed under the terms of the Creative Commons Attribution License, which permits unrestricted use, distribution, and reproduction in any medium, provided the original author and source are credited.
License URL: https://creativecommons.org/licenses/by/3.0/

Keywords: Reading disability, Specific language impairment, Dyslexia, Motor, Imitation, Speed, Dexterity, Comorbidity

Funding: Wellcome Trust Grant no. 053335/Z/98/A This research was supported by a programme grant from the Wellcome Trust (053335/Z/98/A) based at the Department of Experimental Psychology, University of Oxford, and by a Wellcome Trust Principal Research Fellowship awarded to Dorothy Bishop. The funders had no role in study design, data collection and analysis, decision to publish, or preparation of the manuscript.

==============================
Several studies have found evidence of motor deficits in poor readers. There is no obvious reason for motor and literacy skills to go together, and it has been suggested that both deficits could be indicative of an underlying problem with cerebellar function and/or procedural learning. However, the picture is complicated by the fact that reading problems often co-occur with oral language impairments, which have also been linked with motor deficits. This raises the question of whether motor deficits characterise poor readers when language impairment has been accounted for – and vice versa. We considered these questions by assessing motor deficits associated with reading disability (RD) and language impairment (LI). A large community sample provided a subset of 9- to 10-year-olds, selected to oversample children with reading and/or language difficulties, to give 37 children with comorbid LI + RD, 67 children with RD only, 32 children with LI only, and 117 typically-developing (TD) children with neither type of difficulty. These children were given four motor tasks that taxed speed, sequence, and imitation abilities to differing extents. Different patterns of results were found for the four motor tasks. There was no effect of RD or LI on two speeded fingertip tapping tasks, one of which involved sequencing of movements. LI, but not RD, was associated with problems in imitating hand positions and slowed performance on a speeded peg-moving task that required a precision grip. Fine motor deficits in poor readers may be more a function of language impairment than literacy problems.

Introduction

It has been noted for many years that children who are poor readers may also show signs of clumsiness and poor fine motor control. In an early epidemiological study, Rutter & Yule (1970) found an excess of motor impairments in children who were poor readers relative to their IQ (‘specific reading retardation’), regardless of whether this was assessed by parental report, clinical observation or direct assessment. This kind of observation has been used as evidence that reading disability (RD) is not just the result of poor teaching, but has a neurological basis (Ramus, 2004). However, the link between motor impairment and literacy problems remains poorly understood.

One complication is that it remains unclear whether motor impairments are a genuine correlate of RD, or whether they are linked more closely to other problems that co-occur with poor reading. Many children diagnosed with RD (or ‘developmental dyslexia’) also have oral language problems, but these may be overlooked if language is not formally assessed (Bishop & Snowling, 2004). Studies of children with language impairments (LI) provide ample evidence that motor deficits are common in this population. These observations raise two related questions. Will we find evidence of motor impairment if we focus only on poor readers who do not have oral language problems? And if motor deficits are seen in children with combined reading and language impairments, are they the same as those in children who read well despite oral language problems? Because many children have both reading and language difficulties, the existing literatures on RD and LI cannot answer these questions: we need a study of children who have been explicitly assessed for both oral and written language abilities.

Another issue concerns how motor skills are measured. Previous studies have included both fine and gross motor skills, tasks that stress speed versus those stressing precision, and tasks that involve learning versus those that do not. We need to clarify whether RD and LI are associated with distinct types of motor difficulty. The answer to this question has implications for our understanding of possible neurological underpinnings of children’s language and literacy problems.

Where motor deficits have been associated with RD or LI, two types of explanation have been proposed. It could be that the motor deficit co-occurs with other disorders because the causal factors that lead to RD and/or LI are correlated with causal factors that lead to motor problems. Typically this is interpreted at the neurobiological level; for instance, there could be a nonspecific factor, such as delay in myelination, that affects multiple systems at once, or there might be a more specific link, with a deficit affecting a brain region that is involved in both motor co-ordination and language learning, such as the cerebellum. Or the link may go beyond common etiology to involve shared underlying cognitive processes – for instance, language difficulties have been linked to limitations in speed of processing, in sequencing and in imitative capacity – features that are implicated to different extent in different motor tasks. Our focus here is on fine motor skills that might be expected to relate to language impairment, insofar as they share these cognitive characteristics. For instance, theories of language impairment that implicate reduced speed of processing predict there will be links between reduced motor speed and slowed performance on language or literacy tasks that involve rapid processing. Thus, by pinpointing the nature of motor deficits that co-occur with reading or language difficulties, we may cast light on cognitive underpinnings of these disorders, clarifying whether they have similar origins.

We will first review what is known about different fine motor abilities in relation to reading and language impairments and then present new data on a large sample of children assessed for both language and literacy skills. We use the term ‘reading disability’ (RD) rather than ‘dyslexia’ to refer to children with specific problems in reading development, and ‘language impairment’ (LI) to encompass those whose language development is significantly below age level for no apparent reason.

Speed

A number of speeded motor tasks have produced contradictory evidence in individuals with reading difficulties. In some cases, poor readers are reported as slower than peers on tasks such as peg-moving (Fawcett & Nicolson, 1995; Francks et al., 2003; Stoodley & Stein, 2006), bead-threading (Fawcett & Nicolson, 1995; Ramus, Pidgeon & Frith, 2003), foot-tapping (Fawcett & Nicolson, 1999) and finger-tapping (Morris et al., 1998). Fawcett & Nicolson (1999) interpreted these findings as consistent with their theory of cerebellar impairment in RD, as cerebellar patients show similar deficits in these tasks. However, other work has shown that children with RD perform no differently to their peers on speeded tasks including peg-moving (Irannejad & Savage, 2012; Wimmer, Mayringer & Landerl, 1998), bead-threading (Irannejad & Savage, 2012; Savage & Frederickson, 2006; White et al., 2006) foot-tapping (Gaysina, Maughan & Richards, 2010) and speeded writing (Savage & Frederickson, 2006). In addition, Ramus, Pidgeon & Frith (2003) attributed the slowed bead-threading in their study to comorbid Attention Deficit Hyperactivity Disorder (ADHD) or Developmental Co-ordination Disorder (DCD).

In a review of motor skills in LI, Hill (2001) noted that deficits were usually found on speeded motor tasks. An early demonstration of this was by Bishop & Edmundson (1987), who suggested that motor speed might be a marker of neurodevelopmental maturity. They found that on a peg-moving task many 4-year-olds with LIs improved from the impaired to the normal range over an 18-month follow-up period, with a close parallel between improvement in language skills and motor speed. They suggested a possible maturational lag in language impaired children, where the duration of the lag is related to the severity of LI. Bishop (2002) replicated the finding of slower peg-moving in an older group of LI children, and also demonstrated deficits on a simple task that involved tapping a tally counter with the thumb as quickly as possible. Hill (2001) suggested that slow motor performance might be part of a more general slowing of cognitive processing, which has been proposed to affect LI across several modalities (Kail, 1994).

Sequencing

Advocates of the cerebellar theory of RD have noted impairments of sequencing in individuals with RD (Nicolson & Fawcett, 2007). Consistent with this, Stoodley, Harrison & Stein (2006) found that implicit motor learning was poor in adults with RD: on a serial reaction time task, their speed did not improve when the sequence of stimuli was repeated, whereas controls showed implicit learning. In a similar vein, an underlying deficit in the learning of serial-order information has been described in RD, on the basis of impaired Hebbian learning (Szmalec et al., 2011). The Hebb tasks involved implicit learning of the sequence of perceived stimuli, rather than motor sequencing. However, if this kind of learning was impaired in LI or RD, it could lead to problems in automatizing the sequence of movements involved in motor tasks. The finger to thumb task, which involves a repetitive sequence of hand movements, was performed more slowly by children with RD in one study (Ramus, Pidgeon & Frith, 2003). However, as with the bead threading task, the authors suggest this may be due to comorbidity with other developmental disorders. A further study found that children with RD performed as well as peers on the finger to thumb task (White et al., 2006).

The ability to perform a sequence of actions has also been studied in children with LI. Bishop & Edmundson (1987) noted that children with LI made more sequence errors in peg-moving than controls; picking up pegs in the wrong order, or placing them in the wrong hole. Hill, Bishop & Nimmo-Smith (1998) interpreted greater errors in representational gesture production as an inability to implement the precise sequence of movements in children with LI. More recently, several studies have demonstrated impairments of implicit motor learning on the serial reaction time task in children with LI (Gabriel et al., 2013; Hsu & Bishop, 2013; Lum, Gelgic & Conti-Ramsden, 2010; Lum et al., 2012; Mayor-Dubois et al., 2012; Tomblin, Mainela-Arnold & Zhang, 2007).

These studies were prompted by the procedural deficit hypothesis of Ullman & Pierpont (2005) who suggested that children with LI have abnormalities in the procedural memory system, affecting the ability to learn both linguistic and non-linguistic sequences. Nicolson & Fawcett (2007) took this idea further, suggesting that RD and LI might be caused by impairments in different parts of the procedural learning system, with the cortico-cerebellar system implicated in RD, and the cortico-striatal system in LI. However, no studies have directly compared children with these two disorders on the same task.

Imitation

Some tests of motor skill involve copying either a complex hand posture, or a sequence of postures. Problems with motor imitation are usually thought of as characterising autistic disorder, where they are seen as part of a more general problem in social cognition (Williams et al., 2001). However, given that imitation is a key ingredient in language learning, it is worth considering whether children with LI might also have problems with imitating, even in nonverbal contexts. A study by Vukovic, Vukovic & Stojanovik (2010) suggested this may be the case. They asked children to imitate simple and complex movements, with fingers, hands, and arms. Children with LI were able to imitate significantly fewer movements than typically developing (TD) children, showing a marked impairment even for simple movements, whereas control children performed at ceiling levels. Consistent with this was a study by Dohmen, Chiat & Roy (2013), who found deficits in imitation of non-instrumental movements by much younger language-delayed children aged from 2 to 3 years.

In contrast, Hill (1998) found that when asked to copy meaningless hand postures and sequences, children with DCD or LI performed as well as peers, though interpretation of this result was complicated by ceiling effects. On other tasks, Hill (1998) found difficulties in production of representational gestures even when no imitation is required. When producing representational gestures of familiar motor acts, children with LI and children with DCD made more errors than age-matched children, and performed at a similar level to TD children who were 4 years younger; however, this was found regardless of whether the child had to imitate the gesture, or generate it from verbal command. This suggests that difficulties on imitation tasks may be compounded by problems in conceiving and executing motor commands to produce specific manual configurations. Hill (1998) concluded that when performing familiar actions, kinaesthetic information may be especially important, and she suggested that the difficulties of children with LI and those with DCD may have kinaesthetic origins.

Current study

Our first question is whether motor deficits are associated with RD in children who do not have additional LI. We compared children with RD and those with LI to TD children on motor tests that varied in the demands they placed on speed, sequential ordering and imitation. No other study has looked closely at the motor abilities of these two groups on the same tasks. Previous research leads us to hypothesise that, regardless of whether they have additional RD, children with LI will be impaired on tests of speeded motor movements (Bishop, 2002), peg-moving (Bishop & Edmundson, 1987), and motor imitation (Vukovic, Vukovic & Stojanovik, 2010). Our hypothesis is that previous associations with RD on some of these tasks may be due to inclusion of children with LI, and that deficits should therefore not be seen in children with RD.

Second, we ask what kinds of motor skills are most closely linked with reading and/or language abilities in the sample as a whole. We examined correlations between quantitative measures of speech, language and reading skills. This is an exploratory analysis that takes advantage of the fact that we have a wide range of language, literacy and motor measures on a sample of twins, and so can identify correlations that replicate in subsamples that take each member of the twin pair separately. The aim of this analysis was to throw further light on the nature of shared mechanisms between motor skills and language/literacy skills.

Method

Data collection conformed to the Declaration of Helsinki, and ethics approval was obtained from Oxford University’s Experimental Psychology Research Ethics Committee. Parents of participating children gave informed consent, and children gave verbal assent, as agreed by the Ethics Committee. Children were seen in a quiet room at home or school by a trained research assistant. Motor tasks were interleaved within a battery of language and reading ability tests, in a session lasting no longer than 2 h.

Participants

The initial sample included 458 same-sex twins aged 9 to 10 years, recruited through the Twins Early Development Study (TEDS), a non-clinical sample drawn from the general population of twins born in England and Wales (Trouton, Spinath & Plomin, 1994). The selection and categorisation of this particular subsample has been described in detail by Bishop et al. (2009). All children were from White, English-speaking families. As previously described, we oversampled children who had been identified as having difficulties in language or literacy on previous waves of testing, so the numbers of impaired children in this sample was higher than would be found in the general population. We excluded from the sample those with low nonverbal ability (Block Design < 80, N = 30) or high nonverbal ability (all 31 children with Block Design greater than 120 were excluded, plus 47 children with Block Design greater than 114 who would have been in the TD group: This was done to reduce differences between groups in nonverbal ability). In addition, children were excluded on the basis of failing a hearing screen (N = 22), medical conditions (N = 2), evidence of autistic spectrum disorder (N = 6), social deprivation (N = 31 children from a subsample studied by Trzesniewski et al., 2006), being cotwin of a child with major exclusionary condition (N = 13), or failure to complete the test battery (N = 23). This left 253 participants who were aged 9 or 10 years at the time of testing (age M = 9.57 yr, SD = .38).

The criteria used to categorise children were selected to be similar to those adopted by Catts et al. (2005). Children were first grouped according to reading ability. Children were classified as having RD if their average score on two subtests from the Test of Word Reading Efficiency (TOWRE; Torgesen, Wagner & Rashotte, 1999) was below the 13th percentile. Simulations of normal random data showed that assuming a correlation between the two subtests of around .75, this cutoff will select around 11–12% of the population. Children were also categorised according to language ability, either as language typical or language impaired (LI). Where a child had at least two scores more than 1.33 SD below the normative mean on five core language measures (see below for details), they were categorised as LI. Assuming a correlation between the language measures of around .5, this would select around 11% of the population. Mean scores on the tests used to categorise children, and numbers in each group, are shown in Table 1.

Table 1 Means (SDs) on selection and background variables for four groups.

Test	Group	TD	RD	LI	LI + RD	Anova output	
N	117	67	32	37		
% male	40	49	56	68		
Nonverbal ability							
WASI block design	Mean	97.8	99.1	95.3	96.9	F(3, 246.6) = 0.9	
	SD	11.87	11.93	10.99	11.57	p = .439	
Language							
WASI vocabulary	Mean	98.2a	93.3b	83.0c	78.9c	F(3, 223.7) = 33.37	
	SD	13.17	13.05	11.53	12.39	p < .001	
WJ understanding directions	Mean	99.6a	95.9a	78.2b	83.9b	F(3, 246.9) = 31.56	
	SD	13.57	13.62	12.88	13.35	p < .001	
ERRNI comprehension	Mean	98.6a	98.8a	91.8b	88.0b	F(3, 242) = 6.87	
	SD	14.55	14.65	14.13	14.48	p < .001	
ERRNI MLU	Mean	102.1a	97.6a	89.5b	87.8b	F(3, 224.1) = 11.53	
	SD	15.47	15.72	15.2	15.22	p < .001	
NEPSY sentence repetition	Mean	97.1a	92.0b	81.1c	74.7d	F(3, 243.6) = 38.66	
	SD	13.12	13.16	12.01	12.71	p < .001	
Reading							
TOWRE word reading	Mean	102.6a	71.4c	97.6b	68.9c	F(3, 242.8) = 150.84	
	SD	11.54	11.61	11.2	11.48	p < .001	
TOWRE phonemic decoding	Mean	101.2a	75.0c	95.3b	73.1c	F(3, 247.1) = 114.37	
	SD	11.06	11.15	10.53	11.1	p < .001	
Family background							
SES index	Mean	−0.01	−0.06	−0.24	−0.21	F(3, 252) = 1.36	
	SD	0.710	0.710	0.584	0.693	p = .255	
Notes.

TD Typically developing

RD Reading disabled

LI Language impaired

Means with different superscripts differ significantly at the .05 level on LSD test after adjustment of degrees of freedom for twin as random factor.

An index of socio-economic status was available for 91% of the twin pairs, using information gathered when families were first recruited to the Twins Early Development Study (Petrill et al., 2004). This was the sum of z-scores derived from parental educational and occupational status and age of mother at birth of eldest child, and had a mean of 0.10 and standard deviation of 0.72 in the whole TEDS sample. Missing values on this variable were imputed with the sample mean.

Language and reading tasks

Core diagnostic tests

The battery of five core language tests, used to define LI, included expressive and receptive tests of vocabulary and sentence processing: (1) Vocabulary was measured using the Vocabulary subtest from the Wechsler Abbreviated Scale of Intelligence (WASI: Wechsler, 1999); (2) The Understanding Directions subtest from the Woodcock-Johnson III (Woodcock, McGrew & Mather, 2001) measured ability to carry out complex instructions; (3) The Comprehension subtest from Expressive, Receptive and Recall of Narrative Instrument (ERRNI; Bishop, 2004) measured ability to understand questions about a narrative; (4) Mean Length of Utterance from the ERRNI was used as a measure of expressive syntactic complexity; (5) Sentence Repetition from the NEPSY (Korkman, Kirk & Kemp, 1998) was used to assess ability to repeat meaningful sentences of increasing length. Reading was assessed using the TOWRE Phonological Decoding Efficiency and Word Reading Efficiency subtests (Torgesen, Wagner & Rashotte, 1999). These assess speeded reading of real words and nonwords. Scores on the two reading subtests are highly correlated, and were averaged.

Supplementary language and literacy tests

Two additional subtests from the NEPSY, Oromotor Skills and Nonword Repetition were used to assess speech production and phonological memory respectively (Korkman, Kirk & Kemp, 1998). Rapid naming was assessed using an average score from the Pictures and Digits Rapid Serial Naming subtests of the Phonological Assessment Battery (Frederickson, Frith & Reason, 1997). Scores for Reading Accuracy, Comprehension and Rate were obtained from a shortened version of the Neale Analysis of Reading Ability (Neale, 1997), which assesses reading of meaningful texts.

Nonverbal ability

The Block Design subtest from the WASI was administered as a measure of nonverbal ability (Wechsler, 1999).

All tests are standardized, but scores were restandardized to a mean 100 and standard deviation of 15 relative to a normative set of twins who were representative of the whole population, to ensure comparability of norms across tests (see Bishop et al., 2009 for further details and for information on reliability of measures).

Motor tasks

NEPSY Repetitive Fingertip Tapping (Korkman, Kirk & Kemp, 1998) was included as a simple measure of motor speed, which places few demands on sequencing or imitation. Children were required to tap their index finger to their thumb on the same hand, making a circular shape. The experimenter demonstrated, and children were instructed to repeat this action as fast as possible. The time was noted for 32 correct taps. This procedure was administered using the child’s preferred hand, and then repeated with the non-preferred hand. The mean time for 32 taps was inverted to give taps per second, so that proficient performance corresponded to a high score.

NEPSY Sequential Fingertip Tapping (Korkman, Kirk & Kemp, 1998) involves both speed and sequential movement, but places few demands on imitation and does not require such fine dexterity as a peg-moving task. Children sequentially tapped their thumb to each finger of the same hand, from index to little finger. Participants were asked to repeat this sequence as fast as possible, and timed for 8 correct sequences. They first completed the sequences with their preferred hand, and then their non-preferred hand. The mean time for eight sequences was inverted to give sequences per second, so that proficient performance corresponded to a high score.

The Purdue Pegboard is a test that emphasises speed. It involves fine manipulative dexterity under time pressure. It was administered according to the procedure described by Tiffin (1968). Children were given 30 s to move as many small pegs as possible from a well into individual peg holes (in a top-to-bottom line). This task was selected to assess precision grip, which is known to depend on cerebellar activity (Monzée, Drew & Smith, 2004). Participants completed the task twice with their preferred hand, then their non-preferred hand, giving a total of 4 trials. The measure is the total number of pegs placed in holes.

NEPSY Imitating Hand Positions (Korkman, Kirk & Kemp, 1998) assesses the ability to imitate hand and finger positions. Although there is a time limit on the test, the emphasis is on accuracy rather than speed. Children were instructed to copy hand positions demonstrated by the experimenter. A maximum of 20 s was allowed for each of the 12 hand positions. One point was awarded for each correct hand position within the time limit. Again children first completed the task with their preferred hand, and then with their non-preferred hand.

Analytic approach

Previous research has not found reliable effects of language or literacy on difference in skill of the two hands (Bishop, 1990; Bishop, 2001), and so scores for preferred and non-preferred hands were combined to form a composite score for each motor task. Scores were inspected and transformations applied if necessary to correct for non-normality. A natural log transform was used for the two NEPSY Fingertip Tapping tasks, and a rank transform for NEPSY Imitating Hand Positions.

Our primary goal was to consider how language and reading status affected motor performance on the different tasks, and so we included the binary categories of RD and LI as fixed effects in SPSS multilevel linear models for each motor task. The interaction between LI and RD was also tested to see whether the combination of both conditions had a greater impact than would be predicted from their separate effects. Sex was included as a covariate in the model to ensure that group differences were not attributable to this potential confounder, given that some previous studies have found sex differences in RD (Rutter et al., 2004) and a trend for more males was seen in the RD + LI group (see Table 1). Multilevel modelling allows one to conduct analyses that are analogous to conventional analysis of variance, but has greater flexibility. In particular, because our participants were twins, the individual observations were not independent. This was taken into account by including family membership as a random effect in the multilevel models (Kenny, Kashy & Cook, 2006). Effect sizes for main effects are reported as Cohen’s d, based on difference in estimated marginal means divided by the pooled standard deviation. The SPSS script for the analysis is provided in Table S1, together with more detailed explanation.

Analysis of RD and LI effects allows us to relate results to the prior literature, but these categories involve arbitrary subdivisions of continuous scales of language and reading ability. To explore the data in a more quantitative fashion, two-tailed Pearson correlations were computed for language and reading task standard scores with transformed motor scores, for supplementary as well as core diagnostic tests. Because of the large number of correlations computed, there is a risk of finding spurious associations, but the twin design of our study allowed for a natural replication study. Twins from each family were assigned randomly into twin group 1 or twin group 2 and correlations were run separately for each twin group. A correlation was regarded as replicable if it was statistically significant in both twin subsamples. Consistent with previous literature (Smits-Engelsman & Hill, 2012), some motor measures were significantly correlated with nonverbal ability, and so the effect of nonverbal ability was partialled out to ensure than any significant associations were specific to language, and not attributable to general developmental level.

Results

Means for each subgroup on the selection variables, nonverbal ability and SES are shown in Table 1.

Multilevel modelling

Figure 1 shows mean raw scores on the four motor tests in relation to language and reading impairment. Log- or rank-transformed scores, as described above, were used in the analysis where appropriate to improve normality. F-ratios for the fixed effects and interaction are shown in Table 2.

Figure 1 Mean scores on four motor tasks.

Error bars show standard errors.

Table 2 Statistics for main effects and interaction of LI/RD status on four motor tasks.

Effect	Statistic	Finger
tapping	Finger
sequences	Purdue
pegboard	Imitation of hand
positions	
LI	F	0.07	0.06	5.85*	6.42*	
	DF	1,246.6	1,245.9	1,247.8	1,238.8	
	p	.796	.812	.016	.012	
	Cohen’s d	.034	.030	.316	.318	
RD	F	0.02	3.0	0.92	0.48	
	DF	1,247.4	1,247.8	1,245.8	1,247.0	
	p	.900	.084	.338	.488	
	Cohen’s d	.017	.208	.116	.082	
LI × RD	F	1.91	0.05	0.11	0.03	
	DF	1,226.2	1,224.0	1,230.9	1,209.4	
	p	.169	.830	.736	.874	
Sex	F	2.78	0.57	0.56	2.49	
	DF	1,152.1	1,153.04	1,148.5	1,151.8	
	p	.098	.452	.454	.116	
Notes.

* Denotes p < .05.

Different patterns of results were found for the four motor tasks. On the NEPSY Repetitive Fingertip Tapping and Sequential Fingertip Tapping tasks, there was no significant effect of LI or RD, and no interaction between these factors. In contrast, on the Purdue Pegboard and NEPSY Imitating Hand Positions test there was a significant effect of LI. The effect of RD was not significant and there was no interaction between the two conditions.

Correlations

Figure 2 shows the correlations between cognitive tests and motor tests after partialling out Block Design. Results for the two subsamples of twins (each containing one member of a twin pair, selected at random) are shown separately. The full sample was used for this analysis. For a sample of this size, a correlation of .17 is significant at .05 level, a correlation of .23 is significant at .01 level, and a correlation of .29 is significant at .001 level. None of the correlations with finger-tapping were consistently found in both samples at the .05 level.

Figure 2 Correlations of four motor tasks with measures of language and literacy.

Block design (nonverbal ability) has been partialled out. Correlations extending beyond the bold line are significant at p < .05. Those extending beyond the dotted line are significant at p < .01.

The NEPSY Sequential Fingertip Tapping task had consistent, though modest, correlations with speeded reading (TOWRE average) and the NARA subtests, as well as with Sentence Repetition. For this task, the highest correlation in both subsamples was with NEPSY Oromotor Sequences, suggesting that there may be a common core involvement of motor systems in sequencing speech and finger movements.

The Purdue Pegboard task was reliably correlated with Rapid Serial Naming, but correlations with individual language tasks were mostly inconsistent from twin to twin. NEPSY Imitating Hand Positions also showed an inconsistent pattern of correlations in the two subsamples of twins. Only WASI Vocabulary was consistently significantly correlated with this test in both subsamples.

Discussion

Our first question was whether motor deficits are associated with RD in children who do not have additional LI. A large sample of twin children was divided into those with RD, those with LI, those with RD + LI and those with no language or literacy problems (TD). When these four groups were compared on performance on four motor tests, we found that LI status rather than RD status was associated with poor performance on two measures. This suggested that associations between motor impairments and RD may be largely driven by comorbid language difficulties. Furthermore, motor tasks show different patterns of association with LI. This leads to our second question: whether some specific aspects of motor function are linked with language difficulties. We will consider the results in terms of the extent to which motor tasks stressed speed, sequencing and imitation.

Speed

Three of the motor tasks stressed speed: NEPSY Repetitive Fingertip Tapping, NEPSY Sequential Fingertip Tapping and the Purdue Pegboard. The simplest of these tasks, Repetitive Fingertip Tapping, did not discriminate groups: children with RD or LI were as fast as TD children on this measure. This contrasts with a previous study by Bishop (2002), who found reduced speed on a thumb-tapping task in LI children. However, that task involved repeatedly depressing the switch on a tally counter, a novel movement which some children found difficult to do with one hand. Our current data show that if the task demands are reduced to the bare minimum, children with developmental disorders of language and reading can perform as fast as other children.

When the child had to sustain a repetitive sequence of finger movements, there was no main effect of RD or LI in the categorical analysis. However, a correlational analysis on the whole sample revealed reliable associations with the TOWRE measure of speeded reading, and also with the three indices from the Neale Analysis of Reading Ability. This test also showed significant associations with Sentence Repetition and Oromotor Sequences. These correlations were all modest in size, and overall, children with RD did not do more poorly on sequential finger movements than TD children of comparable nonverbal ability and social background.

The Purdue Pegboard, which involved quickly picking up and placing small metal components with a precision grip showed deficits in children with LI. This finding is compatible with previous research that has found that peg-moving performance is impaired in children with LI (Bishop & Edmundson, 1987). Nevertheless, the effect size was small, and no overall association between pegmoving and core language skills was found when the entire range of ability was considered, and nonverbal ability was controlled for.

Deficits were not seen on the simplest speeded motor task in either group, and the LI group showed evidence of motor deficits only when the child was required to perform more intricate movements as fast as possible. Overall, the results do not support an account of generally slowed processing in RD or LI. Rather, it seems that for children with LI, adding time pressure to a task may reveal underlying difficulties with fine motor movements.

Sequencing

Problems in sequencing motor movements have been observed in children with LI doing peg-moving (Bishop & Edmundson, 1987) and gesture production (Hill, Bishop & Nimmo-Smith, 1998), and impaired sequence learning has been observed in serial reaction time tasks in both RD (Stoodley, Harrison & Stein, 2006) and LI (e.g., Tomblin, Mainela-Arnold & Zhang, 2007). In the current study, the one task that involved explicitly producing a sequence of motor movements, NEPSY Sequential Fingertip Tapping, did not show a deficit in either RD or LI. Note, however, that the NEPSY Sequential Fingertip Tapping task is very simple, and the sequence of movements is predictable. Furthermore, the correlational analysis revealed that this motor task was associated with a measure of Oromotor Sequences (repeatedly saying tongue-twisters). This task had not been included in the diagnostic battery for LI, because it stresses articulation rather than language ability. This result suggests that there may be overlap in neural systems involved in programming finger movements and programming articulatory gestures, as has been previously suggested (Bishop, 2002). This suggests it may be important to distinguish between the physical act of producing speech and cognitive aspects of language function when looking for links with motor skill.

Imitation

Imitation tasks have shown that LI children successfully imitate fewer movements than peers (Vukovic, Vukovic & Stojanovik, 2010), though for one study this was only true for familiar gestures (Hill, 1998). The current study confirmed that language impaired children correctly imitated fewer hand positions, despite the fact that most of these were novel gestures. We are aware of no previous research on imitation abilities of children with RD, which was not associated with impaired imitation in the current study.

The interesting question raised by the imitation task is whether there is some supramodal imitation ability that affects children’s ability to learn language as well as their ability to imitate gestures. Imitation involves perceiving a signal produced by another person and then translating that observed percept into a motor programme for producing the same movement. Without imitation ability, language could not be learned. Insofar as imitation has been an explicit focus of research attention, this has mainly concerned children with autism, rather than LI. Deficits in imitation are a hallmark of autism, and, in young autistic children, are predictive of receptive language outcome (Charman et al., 2003). Our results suggest that milder imitative difficulties may underlie slow learning in some children with LI.

Some neurological data support the link between language and imitation. Repetitive transcranial magnetic stimulation (rTMS) to Broca’s area, well known for its role in speech production, interfered with imitation of action (Heiser et al., 2003). The stimulation did not significantly impair production of the same action when the cue to perform was spatial. This specific deficit in action imitation during rTMS suggests that certain parts of Broca’s area have a role in action imitation. MRI has shown functional and structural abnormality in children with LI. Badcock et al. (2012) found reduced activation in Broca’s area in children with LI during an inner speech task, and increased grey matter in this area compared to unaffected siblings and controls. We can therefore speculate that the link between motor imitation deficits and LI reflects developmental abnormality of Broca’s area. This would fit with fMRI data showing that action observation caused activation in Broca’s area (Fadiga et al., 2006). Heiser et al. (2003) described Broca’s area as an area of shared neural mechanisms for communication; through language, action imitation, and action recognition.

Nevertheless, we need to be cautious in interpreting our results. When we considered correlations on individual tests across the full range of ability, the only language test to reliably relate to imitation was WASI Vocabulary, and the effect size was small. Other language measures showed inconsistent correlations with the imitation tasks in the two subsets of twins. Three of the measures, NEPSY Oromotor Sequences, Nonword Repetition and Sentence Repetition, involved explicit imitation of speech, yet none of these subtests was associated with the motor imitation task in both subsets of twins.

Overall, our study confirmed previous work showing a link between imitation and LI. It was interesting to note that this related to vocabulary level, but not to measures that required accurate production of speech sounds. The association was small but intriguing in that it is compatible with neuropsychological studies suggesting a common link between imitation of actions and generation of language.

Conclusions

Our results suggest three reasons for inconsistencies in the literature on motor skills and RD. First, motor tasks tap different aspects of motor function that can be dissociated. We drew a broad distinction between speed, sequencing and imitation, but we used existing standardized tests, which are not designed to tease apart the individual skills that may be contributing to lower performance. For instance, the finger sequencing task was scored according to the speed with which children completed 8 sequences. This measure alone cannot tell us whether some children obtained lower scores because they made sequence errors, or because they were simply slower but accurate. Similarly, deficits on peg-moving might involve dexterity or sequencing as well as speed. Time pressure did not appear to be a major factor affecting performance in the test of imitating hand positions, but nevertheless there was a time limit for each trial, and in future studies it would be worth noting whether some children continued to attempt the posture after the limit expired. In future work it would be useful to devise tasks which are designed to separate the requirements for imitation, sequence and speed, and also to focus on motor tasks that are known to depend on specific motor systems. For instance, it would be of interest to identify tasks that involve cortico-striatal versus cortico-cerebellar systems, and to look more directly at motor learning as well as performance.

A second point is that such associations as exist between motor difficulties and language/literacy problems are small in magnitude, especially when potential confounders have been accounted for. The largest correlations between motor and language/literacy measures in this sample were below .4, and the significant effect sizes seen in Table 2 were around .3. Such effects are not easy to detect, especially in small samples, and may vary from sample to sample, as is evident from the correlational analysis.

A third conclusion from this study is that RD and LI often co-occur, and motor impairments that are seen in poor readers may be more a function of their LI than their literacy problems per se. We did not examine other comorbidities, such as attentional problems that often co-occur with both reading and language impairments, but there is some evidence that these too can be a factor affecting whether or not motor impairments are observed (Raberger & Wimmer, 2003; Ramus, Pidgeon & Frith, 2003). It would be premature to conclude there are no motor impairments in RD, given that our test battery was of necessity limited. Measures of balance, posture and muscle tone were not included in our study, and their involvement in RD has been debated (e.g., Fawcett & Nicolson, 1999; Irannejad & Savage, 2012; Needle, Fawcett & Nicolson, 2006; Rochelle & Talcott, 2006). However, the distinctive patterns of associated motor impairment obtained here suggest we will obtain more coherent results if we assess both oral language and literacy skills when looking for neurobiological bases of these developmental disorders. Where RD occurs in the absence of other comorbidities, motor difficulties are unlikely to be found on tests that stress speed and dexterity of hand function.

Although we did not find convincing evidence of links between RD and motor skill, once language had been taken into account, we did find significant associations with LI and performance on a peg-moving task that stressed fine motor dexterity, and on imitation of hand postures. These effects were not large, and were unlikely to be of practical importance for most children. Nevertheless, such comorbidities are especially intriguing when they involve skills that do not, on the surface, appear to have much in common. They may indicate common causes for motor and language difficulties, which could give clues to etiology. For instance, the association between problems with nonspeech oral movements and language difficulties in people with a mutation of the FOXP2 gene has pointed to a role of this gene in the development of cortico-basal ganglia circuits which, in turn, have generated a rich body of research using animal models (Enard, 2011). The etiology of common language and literacy problems is seldom as straightforward as this, but by studying comorbid difficulties, we may uncover underlying pathways that are implicated in speech, language and motor skill.

Supplemental Information

Supplemental Information 1 Consent form used for parents of participants

Click here for additional data file.

Table S1 SPSS script for mixed models analysis of dependent variable Purdue_raw, with factors LI and RD, and sex as covariate

Click here for additional data file.

We thank the twins and their families and teachers who participated in this research. This study would not have been possible without generous assistance of Robert Plomin, Bonamy Oliver, Alexandra Trouton, and other staff from the Twins Early Development Study.

Additional Information and Declarations

Competing Interests

Author Contributions

Human Ethics

The authors declare they have no competing interests.

Annie Brookman analyzed the data, wrote the paper.

Sarah McDonald and David McDonald performed the experiments, critique of ms.

Dorothy V.M. Bishop conceived and designed the experiments, analyzed the data, wrote the paper.

The following information was supplied relating to ethical approvals (i.e., approving body and any reference numbers):

Department of Experimental Psychology Ethics Committee, University of Oxford. Approval by letter. (This committee is now superseded).

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
