# Peer review of "Fine motor deficits in reading disability and language impairment: same or different?"

_PeerJ, doi:10.7717/peerj.217_

## Round 0.1 · original submission · Major Revisions

Motor deficits in reading disability and language impairment: same or different.
A Brookman, S McDonald, D McDonald, D Bishop
EP, Oxford.

Dear Dorothy,

Thank you for submitting the manuscript entitled “Motor deficits in reading disability and language impairment: same or different” to PeerJ. This has been assessed by two of your suggested reviewers. I have also reviewed the paper myself, as an endeavour toward “paragon editorship” ☺. This being the case, I am somewhat reluctant to admit that my own review of the manuscript very closely matches that of Rob Savage, and so am very tempted to say “do what he said”. However, in an effort to provide some direction about priority, I would suggest that, at the very least, the manuscript needs to provide (1) a more thorough and balanced review of the literature that includes research that does not necessarily support a link between RD and motor deficits; (2) clearer justification for choice of motor tasks, and an earlier explanation for what is task is supposed to measure (ie speed, sequencing, imitation; in the Introduction or Methods – not the Discussion); (3) clearer description (including descriptive data) that illustrates the nature of the groups within the sample; (4) a clearer description of the results; (5) a consideration of individual differences (subsets) of children; and (6) a more convincing justification of why the results are perceived as support for a link between RD and LI and motor deficits when the correlations are so low and inconsistent.

Another top priority is improving the clarity of the manuscript in various places. Rob provides a number of specific suggestions about how this might be done. In addition to these suggestions, I outlined aspects of the manuscript that I found a bit confusing, and offer suggestions about how these areas might be clarified. I hope you find the suggestions of both Rob and myself helpful in further developing your manuscript. (Note: You will notice that I have not referred to a second reviewer. This is because their review was overwhelming positive - which has been fully considered in this editorial decision).

Introduction

(1) The introduction needs tightening up a bit. There needs to be a more logical flow between paragraphs, which often jump from one topic to another. For example, paragraph three jumps from discussing how motor skills are measured, to how RD and LI are defined in the current study (paragraph 4), to the questions asked by the study (paragraph 5). The information in paragraph 4 feels like a late addition, and may perhaps be better placed in the methods.

(2) The questions addressed by the study are not clear. At one point of the Introduction it is said “Will we find evidence of motor impairment if we focus on poor readers who do not have oral language problems? And if motor deficits are seen in children with combined reading and language impairments are they the same as those in children who read well despite or al language problems?”. At a later point it is said “Our first question is whether motor deficits are associated with both RD and LI” and “Second, we can ask what kinds of motor impairment are most closely linked with reading and/or language problems”. And there was no clarification of the true questions of the study at the end of the Introduction.

While these questions are related, they are not the same. And it is never clear exactly what issue the study is tackling. The manuscript needs to pinpoint exactly what questions are being addressed, and offer hypotheses (based on previous research) about what the study should find. If no such hypotheses can be made (because there is no previous research), then it needs to be clarified that the analyses will be exploratory in the effort to create new hypotheses to be tested by future studies. The fact that the study (cleverly) has two twin groups (I think this is a real strength of the study) means that if you are forced to do exploratory analyses, you could do the exploratory analysis with one group, and the hypothesis-based testing with another. I am not suggesting that this definitely should be done. But it might be something to consider if the aims of the study can only be tested in an exploratory way.

(3) The manuscript uses various different terms for RD and LI (e.g. RD, dyslexia, LI, SLI). Once defined, just stick with RD and LI throughout.

(4) Even if the fourth paragraph of the Intro is moved to the Methods, it needs a bit of clarification. It is not clear to me why this manuscript supposes that RD is different to “developmental dyslexia” (Note: this impression is created by the statement that RD “overlaps substantially” with developmental dyslexia, and “the nomenclature emphasises that this is not clear-cut syndrome with sharp diagnostic boundaries). I must be missing something here because I am not aware of any dyslexia researcher who believes that developmental dyslexia does not fall along a continuum (this includes research who work with very extreme cases with unusual dyslexia profiles), or that we all use arbitrary cutoffs to define our groups. Perhaps you are defending yourself against the approach of someone in particular? If so, you may need to reference this work specifically.

(5) At the end of the Introduction, in the Current Study section, what is needed is a clearer link between the previous research that has been reviewed, the questions that are generated by this research, the aims of the study, and the hypotheses. These aims and hypotheses can then be used to structure the Discussion.

Method

(6) Please clarify what is meant by “community sample”

(7) How is a sample “enriched” by included children with RD and LI. Perhaps a different word could be used?

(8) What is meant by “oversampling” exactly?

(9) Provide numbers of children lost due to low nonverbal IQs as well as hearing loss, autism etc.

(10) Justify why different cut-off points were used to define impairment in reading (13th %ile = -1.15 SD) and language (-1.33 SD = ~9 %ile). Or at least acknowledge that this was the case (Note: the difference is not huge, so this is not a deal breaker).

(11) Was the “TD” group simply composed of all the children in the recruited sample who were not placed in the RD, LI, or LI+RD groups? If this was the case, then not all of these children would necessarily be typical developers since they might have reading scores at the 12th %ile and language scores at -1.32 SD (ie they just fell above the cut-off points). If, in turn, this was the case, it would be important to exclude children from this group whose reading and language scores did not fall within the average range (e.g., only include those with all reading and language scores between -1 SD and +1 SD, for example).

(12) It is not clear why the supplementary tests were included. Remove if not relevant to clearly defined aims and hypotheses.

(13) Under Motor Tasks, explain which aspect of motor skill each task was supposed to measure (ie which tasks were used to measure speed, sequencing, and imitation).

(14) What does “relative hand skill” mean. Do you mean preference?

(15) There is absolutely no justification provided for including Sex as a covariate. If this is, indeed, necessary (which seems unlikely, given inconsistent findings for sex differences in RD, at least), then it needs to be clearly introduced and justified in the Intro (ideally) or Methods.

(16) I really liked the idea of randomly assigning twins to twin group 1 and 2 and running the correlation analysis twice. It’s a great idea in terms of reliability. Is there a particular reason why you couldn’t do the same for the first analysis? That is, for twin group 1, do simple ANOVAs (and posthoc t-tests) comparing the four groups on each motor task. Then do the same for twin group 2? This would be a great way to understand the reliability of results and hence the strength of the final interpretations.

(17) I’m not a strong advocate of correcting for multiple corrections for corrections sakes (ie applying Bonferroni corrections willy-nilly). Interpreting the outcomes of multiple comparisons is much more complicated than that. However, I am strong believer in minimising the chance of spurious results, and a believer in hypothesis testing, since this maximised reliability. Thus, if the correlational analysis is necessary (which is currently unclear because the exact aims/hypotheses of the study are not exactly clear) I would suggest limiting the correlation analysis to only those variables that relate to the hypotheses that are being tested. At the moment, it looks a bit like a fishing expedition. And the strength of the correlations are generally so weak, the results look more like an argument against an association between RD and LI and motor deficits than otherwise.

Discussion

(18) This will need to be revised to address the aims and hypotheses of the study, once they have been clearly defined.

(19) Care will need to be taken to not over-interpret the outcomes. I agree with Rob that, at the moment, the outcomes look more negative than positive in terms of an association between motor deficits and RD and LI. Which is fine. But this is not the interpretation offered in the study. This needs very careful consideration.

(19) Should General Discussion been renamed Conclusions?

·

Basic reporting

I generally found the paper to be well-written and clear in the main text and the main idea behind the paper is sound and transparent to a reader. This is a good issue to explore, and there is mileage in an article on this topic. I hope you find the comments below helpful and useful in revising the paper.
While generally clear, I did have concerns about the reporting of specific aspects of the methodology as I outline below. Beyond this I felt that the literature was much fuller on issues related to language than to literacy difficulties. It also seemed to me that there were areas where citation was very sparse generally (e.g. the first 2 pages of text that could lead quicker to the main questions of interest).
More importantly I felt that the literature did not feature the rather large body of work finding little evidence of motor-RD associations. This is a problem as is because many researchers would find motor (or cerebellar) deficit-RD difficulties a very controversial idea. This could (but need not preferentially) include our own work (e.g. Irannejad & Savage, 2012 Annals of Dyslexia; Savage, 2007 Cognitive Neuropsychology, Savage et al. 2005 Perceptual and Motor Skills, Savage & Frederickson, 2006, Journal of Learning Disabilities Savage et al., 2005 Journal of Learning Disabilities; Savage 2004 Reading and Writing) showing no effects, but also that of many others (e.g. Ramus et al 2003 JCPP; Raberger & Wimmer, 1999, 2003; Wimmer et al (see below) also Needle et al. 2006 European Journal of Cognitive Psychology). There are several systematic reviews finding no reading–motor link either (e.g. Hammill, 2003; Rochelle & Talcott, 2006 JCPP ) and this needs to be considered (see also Hattie’s text, 2002 Visible Learning). Work by White et al. 2006 Developmental Science (see also the replies) and of Raberger & Wimmer, 1999, 2003, Wimmer et al., 1998 (Scientific Studies of Reading), Journal of Learning Disabilities, 1999 and Ramus 2003) deserves special attention as they looked at the issue of comorbidity and found motor deficits on a subset of RD children only when linked to general motor or attention difficulties.

Experimental design

I was not sure that the logic of looking at all aspects of motor processing in both RD and LD contexts was justified. Why explore imitation in RD samples?
I think it important (even crucial) to know how typical this sample are, (even of atypical samples!). Does the fact that they were a sample of twins affect the generalization in any way? While reference to another paper for brevity is generally a good thing, details of that need to be in this manuscript.
We need to know more about the sample generally (how old were they? (maturation has been cited as an explanation of motor problems/ differences). What were their reading experiences?, SES? Were these potentially confounding factors demonstrably equal across groups?
For a factorial design like this, I think we need to first know that the categorization used has produced groups of ‘pure’ LD and RD and a mixed group in relation to the typical children. A table of means SD and effect sizes for all 4 groups on selection measures and analysis showing that groups did indeed differ (and not differ) exactly as anticipated on these measures, (but not presumably on more general measures like non-verbal IQ, SES) are needed here.
I would like to see details of the analysis (e.g. a 2x 4 ANOVA) with F P DF etc as per APA guidelines. The term ‘hierarchical’ is confusing as it can refer to many analyses. For example to large nested designs, I assume this was not the case here. How were effect sizes calculated? In relation to controls? and why none for the RD x LD condition?

Validity of the findings

I wondered about the fact that of 4 motor tests only one showed any effect for the RD alone versus the typical children, and this with a tiny ( arguably practically unimportant ) effect size, especially in contrast to that routinely found in other papers for pseudoword decoding in RD for example. The correlations show weak r (again small effect sizes) for most associations of interest. Crucially, findings could thus alternatively be interpreted as no evidence of a clear motor-RD link, with the proviso that there might however be a link in the sub-set of LD-RD comorbid children!
Just generally, might the effects (in RD or in RD+LD groups) be present only for a subset of such children? Finally, the claim in the abstract that an RD-motor link has been found after LD has been ‘accounted for’ is not borne out in any analyses.
In addition the tests used to identify reading were both speeded tests, these might tap fluency and be more likely to find a reading speed-motor fluency link. RD in opaque orthographies like English is not reliant on fluency but rather is routinely diagnosed based on accuracy. By contrast the LD tasks are all accuracy tasks. In addition, different criteria are used to sample the 2 groups: 13th %ile for RD versus 1.33 SD below mean for LD. (BTW it needs to be clear if this was on all tests, some, an average of 5 etc for the LD condition).

Additional comments

I think that while it a good idea to explore the cerebellar idea for subgroups, how closely tied to that specific idea is the present work? Nicolson and Fawcett place a good deal of weight on a stork balance task (i.e. it is the main one with bead threading in their dyslexia screening test), and there is an additional implicit learning / procedural learning idea in their theorizing (not to mention their early emphasis on dual task and automaticity, muscle tone, posture and a range of other ‘symptoms’ ). Not all these skills were assessed here.
Even if found a motor difficulty need not be causal in reading (or language) difficulties. It might be an epiphenomenon of (e.g.) the sampling used, or behavior-genetic phenotype for RD and/or RD, or a consequence of some even general form. This issue might need to be considered in the interpretation.
Might there be motor-RD link ONLY for children with poor handwriting (dysgraphia), or (most obviously) for a subset with identified developmental co-ordination problems?

·

Basic reporting

No comments

Experimental design

No comments

Validity of the findings

No comments

Additional comments

The current study is a welcome contribution to the field. It is the first study to my knowledge which combines a number of motor tasks which tap sequencing, imitation and speed within the context of children with reading disabiilty, language impairment or both.The design was very sound, the data analysis careful and appropriate and the interpretation was measured and logical.

---

## Round 0.2 · Minor Revisions

Fine motor deficits in reading disability and language impairment: Same or different?
Brookman, McDonald, McDonald, Bishop

Dear Authors,

Thank you for submitting a revised version of your manuscript. I certainly agree that the paper is clearer and the methods and outcomes offer some interesting insights into the relationship between language, reading and motor abilities. It was particularly interesting to note that the re-analysis revealed that motor deficits are more likely to be associated with LI than RD. That is a neat result.

I did not send your manuscript back to the original reviewers since your responses suggested you had addressed almost all of the concerns. Instead, I re-read the manuscript afresh. It is my opinion that if you are able to address all the minor issues outlined below, which all relate to clarity of presentation, the manuscript should be close to publishable.

Introduction

1. I think the Intro reads nicely overall. There were only two places that I struggled. The first was in the section on “Imitation and praxis”. The introduction of the term “praxis” was confusing since it was not defined, and it did not fit into the existing “structure” of motor deficits that had been set up throughout the manuscript (ie in terms of speed, sequencing, and imitation). I think the simplest way forward may be to remove the term “praxis” from the title of this section (in the Discussion too), and slightly reword the second paragraph of this section to something like:

In contrast, Hill (1998) found that when asked to copy meaningless hand postures and sequences, children with DCD or LI performed as well as peers. However, interpretation of this result is complicated by ceiling effects on these tasks, and by the fact that these children made more errors than age-matched children when producing representational gestures of familiar motor acts. Indeed, their performance matched that of TD children four years their junior, regardless of whether gestures were imitated or generated from verbal commands. Hill suggested that the motor difficulties of children with LI and those with DCD may have kinaesthetic origins.

2. The second section that needs some revision is the “Current Study” section. It would be easier to understand the logic if the two paragraphs were merged. It is difficult to explain how this could be done in words, so here is an example of what I mean (note, this paragraph includes some minor rewording and rearrangement to adjust for the comment above):

Our first question is whether motor deficits are associated with RD in children who do not have additional LI. The current study compared children with RD and those with LI to TD children on motor tests that varied in the demands they placed on speed, sequencing, and imitations. No other study has looked closely at the motor abilities of these two groups on the same tasks. However, related research leads us to hypothesise that previous associations with RD on some of these tasks may be due to inclusion of children with LI, and that deficits should therefore not be seen in children with RD. In addition, we hypothesise that children with LI, regardless of whether they have additional RD, will be impaired on tests of speeded motor movements (Bishop, 2002), pegmoving (Bishop & Edmundson, 1987), and motor imitation (Vukovic et al, 2010).

Our second question is what kinds of motor skills are most closely linked with reading
and/or language abilities in the sample as a whole. We addressed this question by examining correlations between quantitative measures of speech, language and reading skills. This is an exploratory analysis that takes advantage of the fact that we have a wide range of language, literacy and motor measures on a sample of twins, and so can identify correlations that replicate in subsamples that take each member of the twin pair separately. The aim of this analysis was to throw further light on the nature of shared mechanisms between motor skills and language/literacy skills.

3. You will notice that the above paragraphs no longer include:

“Both RD and LI are more common in boys than in girls, so to ensure that any associations with motor performance were not just due to poorer motor skills in boys, we used sex as a covariate in all analyses”.

In your response letter, you make it quite clear why you wish to use sex as a covariate (“avoid a potential confound … if boys were general worse at motor skills and more likely to have RD and LI). This is expressed quite clearly. However, the sentence in the manuscript is less clear. In addition, in order for even the clear sentence to hold up, we would need two types of evidence: (1) boys are more likely to have LI and RD than girls, and (2) boys are more likely to have motor deficit than girls. Regarding (1), studies done in the 1990’s that removed recruitment biases from RD samples showed that RD is not more common in boys than girls (ie they screened 100s of children in schools without asking for parental consent, and this showed a similar number of girls and boys have RD). So, in terms of RD, the evidence does not hold up regarding (1). This may not be a big deal, since your study has found that motor deficits are related to LI rather than RD. So, what is the evidence that more boys have LI than girls? And by evidence, I mean evidence that is not biased by referral? Regarding (2) is there any evidence that more boys have poorer motor skills than girls. I’m afraid a clearer more logical justification for controlling for sex in the analysis is still needed. Note, I have no personal preference either way. It is just about presenting a logical argument for readers so they can understand why it was done.

In addition, if such a justification is possible, it would be better located under the Analytic Approach section.

4. In this new version, the terms RD and dyslexia are still used inter-changeably throughout the manuscript, as are LI and language impairment and SLI. To avoid confusion, I suggest that all the various terms are stated in the first page of the manuscript and, unless it is important not to, only RD and LI used throughout the rest of the manuscript.

5. rewrite vs. and versus

6. define TD at first mention (ie line 111 – remove “(TD)” from line 127 and from that point on, just use TD unless inappropriate.

6. In at least one place, the order of refs in parentheses is not alphabetical. Please double check throughout the manuscript.

Methods

Participants

1. There are two bits of information needed for this section. One explaining how, where, and when children were tested. This could potentially be added to paragraph 5 of this section.

2. The other is an explicit statement of the final numbers for each group. This paragraph exists but is in the Results section.

3. The order in which the information is presented in this section is difficult to follow.

4. To address 1 and 2, I would suggest the following. This section currently has five paragraphs. I would suggest making paragraph 5 paragraph 1; then putting paragraph 1 as 2; paragraph 2 as 3; paragraph 4 as 4; and paragraph 3 as 5. I would then add info re: point 1 (immediate above) at the end of the new paragraph 1. And I would place the info re: point 2 (immediate above, which you have in the results) after new paragraph 4.

Language and reading tasks

1. Another way the clarity of manuscript could be improved would be to use consistent terms to refer to what the children were tested for. In the Methods, Results, Discussion, tables and figures, sometimes the measured are defined by the actual test (e.g., TOWRE phonemic coding) and sometimes by the cognitive skill (e.g., reading rate). This makes it difficult for the reader. I would suggest that the “Language and reading tasks section” is used to define the terms that will be used consistently to refer to each test throughout the manuscript, and I suggest that this term refers to the cognitive skill rather than the test, since readers will not necessarily know what, say, Nonword Repetition or WJ Comprehension (Figure 2) measures.

2. Remove “Next” from line 155.

3. Double check that names of tests and subtests are in Title Case.

4. For each test, include a statement of what cognitive skill/s it is supposed to measure, and outlined what scores are produced, and how to interpret them (e.g., what is the mean and what is the SD?)

Motor tasks

1. Remove first paragraph since this information should be put in a new section under Participants (ie the new section that describes when and how the children were tested).

2. What did the Purdue Pegboard test measure: speed? Sequencing? Imitation?

3. What were the scores for the Purdue Pegboard test? How were they interpreted (ie were high scores good, as you have explained for the other tests?)

Analytic approach

1. Line 254. Put new/clearer justification for sex as covariate here – remove from Intro.

2. At this point, also explain why NVIQ was controlled for (this is currently not explained/justified)

3. At the end of the last paragraph, state what criteria were used to decide if the twin correlations were reliable or not.

4. You say in your response letter that you controlled for SES in the analysis, and you explained why you did this. However, I cannot see this in the Analytic Approach section.

Results

1. Move first paragraph to Participants section as outlined above

Correlations

1. Reword Block Design as non-verbal IQ (this is an example of the importance of using terms that refer to cognition rather than the test since some readers will have no idea what Block Designs actually is or what it measures; of course they could refer to the Methods, but this makes understanding the manuscript harder than it needs to be).

2. Justify why non-verbal was partialled out under Analytic Approach

Discussion

1. The first paragraph needs to be rewritten to briefly remind the reader what the questions of the study were, what was done to tackle the questions (who tested on what and how analysed) and then describe what will be done in the Discussion. In this case, you did relate the outcomes to each question in turn as one might predict, but you discussed how the results related to each type of motor skill in turn.

2. After each paragraph within each section, a final concluding sentence is needed that states what the current and previous findings tell us about the questions addressed in this study.

3. The same is true for the different sections (speed, sequencing, imitation). A final concluding paragraph is needed to bring all the evidence together to make comment about the status of motor speed, sequencing, and imitation in kids with RD, LI, and LIRD.

4. It is unclear what the paragraph starting at line 324 is referring to (ie the Sequencing Task). It does not refer to previous findings. And it does not provide a final concluding paragraph to make a statement about what the existing evidence tell us about the performance of RD, LI, or LIRD children on such tasks.

Tables and figures

More information is required to describe any symbols (e.g., *) abbreviations (SD, LI, TD, RD). All numbers should have consistent number of DPs (usually two). Whilst APA format may not be the official format of PeerJ, it provides a good guide about the type and amount of information required for readers to properly interpret information of graphs and in figures. Particularly for young researchers who have not yet encountered many obsessive type-setters.

Response letter

1. It is unusual to comment about a response letter, but you make a good point in it that would benefit the manuscript. Specifically, you note that:

“we were not arguing that these effects were large, nor that they were or practical important: comorbidities are especially intriguing when the involve skills that do not, on the surface, appear to have much in common, and even small associates may be theoretically informative Insofaras there is an association, we have to understand why - it may indicated common causes for motor and language difficulties, which could give clues to etiology.

I think the second last paragraph of the conclusion would greatly benefit from inserting this point. And it would be helpful if you could provide a specific example of how correlations have indicated common causes for skills that do not appear to have much in common, and how this has provided clues about etiology. Or at least how they “might” in the context of the focus of your study. Just to make your point more concrete for readers. I feel this is an important point to make in this particular field of research because it applies to so many studies.

---

## Round 0.3 · Minor Revisions

Dear Dorothy,

Thanks for sending a revised version of the manuscript. It was a pleasure reading it through for a final time. I particularly enjoyed the discussion. While it was (appropriately) cautious, in my opinion, it did a really nice job of explaining why one might spend time looking at motor deficits in children with learning difficulties.

There are only a few very final minor changes before I hit the big green Accept button. It may make life simpler if we deal with these now rather than leave them for the typesetting stage:

1. I understand your point about naming the variables after the tests rather than the cognitive processes. I think my main issue related to consistency of referral to the variables. This is mostly OK now. There are just a couple of hangovers. First, Table 1 refers to WJ Comprehension. But this is not mentioned in the Methods. I think it is called WJ Understand Directions. If this is the correct test, then this needs to be reconciled in some way.

In addition, Table 1 refers to both the test name and subtest for most variables (e.g., ERRNI Comprehension, TOWRE word reading, ERRNI MLU). For ease of understanding, you may want to apply that same convention to all the variables in the table and text (e.g., WASI Block Design). Or you may not. This is not critical – just good for consistency.

2. In the title for Table 1, you refer to LSD and DF. It may help the reader if you write these in full.

3. In Figure 2, you refer to the fourth test as “Imitating hand positions”. Throughout the rest of the manuscript, you refer to it as “Imitation of hand positions”. So you may wish to change this for consistency.

4. There are a couple of missing commas: one on line 46 between the words “development” and “and” (i.e., development, and ‘language impairment’); and one on line 166 after “addition”.

5. There are a couple of “howevers” in the paragraph from line 118 to 129 that keep tripping me up. I think this paragraph would be easier to read if these were removed (lines 120 and 124).

6. I’m not quite sure which “et al.” convention you are using in terms of first referral to multi-authored refs. However, I suspect there are just a few references in the text that you may want to double check since “et al.” is used at first mention even though there are just a few authors. These are at:

line 82 (Szmalec et al.)
line 97 (Maor-Dubois et al.)
line 379 (Charman et al.) – this is also underlined for some reason
line 387 (Badcock et al.)

I hope that all makes sense. When you send the final version back to me, I shall hit the big green Accept button without further review.

Genevieve

---

## Round 0.4 · accepted · Accept

Dear Dorothy,

Thanks for the revisions. We are all good to go.